# Doripenem in the Treatment of Patients with Nosocomial Pneumonia: A Meta-Analysis

**DOI:** 10.3390/jcm11144014

**Published:** 2022-07-11

**Authors:** Chienhsiu Huang, Ihung Chen, Yalun Yang

**Affiliations:** 1Department of Internal Medicine, Dalin Tzu Chi Hospital, Buddhist Tzu Chi Medical Foundation, Chiayi 62247, Taiwan; b89401098@nut.edu.tw; 2Department of Nursing, Dalin Tzu Chi Hospital, Buddhist Tzu Chi Medical Foundation, Chiayi 62247, Taiwan; df132950@tzuchi.com.tw

**Keywords:** doripenem, hospital-acquired pneumonia, ventilator-associated pneumonia, *Pseudomonas aeruginosa*

## Abstract

Introduction: Clinically, doripenem therapy for nosocomial pneumonia remains a serious concern. The purpose of this meta-analysis was to explore the efficacy and the safety of doripenem therapy for nosocomial pneumonia in comparison with other antimicrobial agents. Methods: Studies were eligible for inclusion only if they directly compared the clinical effectiveness of doripenem and other antimicrobial agent therapies for nosocomial pneumonia in adult patients between 1 January 2000 and 30 April 2022. All studies were included if they reported one or more of the following outcomes: clinical cure rate, microbiological cure rate, all-cause mortality, and adverse events. Results: Six randomized controlled trials and three retrospective studies were included in the meta-analysis. There were 952 patients in the doripenem group and 1183 patients in the comparator group. The comparator antimicrobial agents included imipenem/cilastatin, meropenem, and piperacillin/tazobactam. Seven studies had a high risk of bias. Doripenem therapy for nosocomial pneumonia had a microbiological cure rate, a clinical cure rate, an all-cause mortality, and adverse events similar to those of comparators. Conclusions: The efficacy and the safety of doripenem therapy for nosocomial pneumonia were comparable with those of comparators. Randomized controlled trials are needed to confirm the role of doripenem in nosocomial pneumonia therapy.

## 1. Introduction

Doripenem is a carbapenem antibiotic that has a broad spectrum, antibacterial activity similar to imipenem against gram-positive pathogens and an antimicrobial spectrum similar to meropenem against gram-negative organisms [1,2,3]. Doripenem activity against *Pseudomonas aeruginosa* has been reported to be comparable to that of meropenem [4,5]. In the study by Kollef et al., a fixed seven-day course of doripenem was compared with a fixed ten-day course of imipenem/cilastatin for the treatment of ventilator-associated pneumonia (VAP) [6]. The study demonstrated a lower clinical cure rate and a higher mortality rate among doripenem-treated subjects than among imipenem/cilastatin-treated subjects. The clinical trial of doripenem for VAP treatment was terminated due to safety concerns. In March 2014, the US Food and Drug Administration (FDA) approved doripenem label changes to highlight the increased risk of death in VAP patients. The US FDA approved doripenem only for the treatment of complicated intra-abdominal infection, complicated urinary tract infection, and pyelonephritis [7,8]. However, doripenem has been approved in Europe for the treatment of complicated intra-abdominal infection, complicated urinary tract infection, and nosocomial pneumonia, including VAP [9]. Several studies have reported the clinical effectiveness of doripenem therapy for nosocomial pneumonia [10,11]. Therefore, we performed a comprehensive and an updated meta-analysis of the clinical outcomes associated with doripenem therapy for nosocomial pneumonia patients, including VAP and hospital-acquired pneumonia (HAP) patients. We hypothesize that doripenem has clinical efficacy in the treatment of nosocomial pneumonia. The purpose of this meta-analysis was to explore the efficacy and the safety of doripenem in patients with VAP and HAP in comparison with other antimicrobial agents.

## 2. Method

### 2.1. Data Search Strategy

The literature search was performed using PubMed, Web of Science, and Cochrane Library databases to identify all included clinical studies and meta-analyses or systematic reviews on the topic from 1 January 2000. In the databases, we used the following search string: ‘(doripenem OR imipenem OR meropenem OR piperacillin-tazobactam) AND (pneumonia OR nosocomial pneumonia OR ventilator-associated pneumonia OR hospital-acquired pneumonia) AND [in PubMed] (random OR prospective OR retrospective OR cohort OR observational OR clinical trial)’, AND [In Web of Science] (random OR prospective OR retrospective OR cohort OR observational), AND [in Cochrane Library] (random OR prospective OR retrospective OR observational). We examined treatment options, including doripenem, imipenem, meropenem, or piperacillin-tazobactam, for patients with pneumonia and searched relevant articles published from inception to 30 April 2022. We included all clinical studies, including retrospective observational studies, prospective observational studies, and randomized controlled trials. Conference abstracts were also searched. Previously published systematic reviews and meta-analyses were reviewed to identify any additional studies that may have been missed in the primary literature search. No language, publication date, or publication status restrictions were imposed.

### 2.2. Study Selection and Data Extraction

To determine the eligibility of identified trial reports, each study was independently screened and reviewed for eligibility by two authors. After excluding duplicates, two investigators screened the titles and abstracts of all studies retrieved to identify eligible records. After excluding irrelevant studies, all of the relevant articles were reviewed by reading the full texts to determine eligibility. Data regarding author, year of publication, country, study design, pneumonia type, total number of patients receiving doripenem, total number of patients receiving other antimicrobial agents, and administered antibiotics were extracted. Data on the microbiological cure rate, the clinical cure rate, all-cause mortality, and adverse events for patients with nosocomial pneumonia were manually extracted from the eligible full text articles. Any disagreement was subsequently resolved with the consensus of the review team and discussion with a third author.

### 2.3. Inclusion and Exclusion Criteria

Observational studies with inadequate levels of evidence are not as meaningful as RCTs. Because a very small number of RCTs were available, we included all clinical studies, including retrospective observational studies, prospective observational studies, conference abstracts, and randomized controlled trials. The studies were considered eligible for inclusion only if they directly compared the clinical effectiveness of doripenem with those of other antimicrobial agents in the treatment of adult pneumonia. Doripenem was administered at dosages ranging from 250 mg every 12 h to 1.0 gm every 8 h. Meropenem was administered at a dosage of 1.0 g every 8 h. Imipenem was administered at a dosage of 500 mg every 6 h or 1.0 g every 8 h. Piperacillin-tazobactam was administered at a dosage of 4.5 g every 6 h. All studies were included if they reported one or more of the following outcomes: clinical cure rate, microbiological cure rate, all-cause mortality, adverse events, Pseudomonas aeruginosa (PA) pneumonia clinical cure rate, PA pneumonia microbiological cure rate, and PA pneumonia all-cause mortality. Studies with a population of participants who were younger than 18 years were excluded.

### 2.4. Definitions and Outcomes

The primary outcome was all-cause mortality. All-cause mortality was the death rate from all causes of death for a population in a given time period. Secondary outcomes were the clinical cure rate, the microbiological cure rate, and adverse events. Clinical cure was defined as resolution of clinical signs and symptoms of pneumonia at the end of treatment. Microbiological cure was defined as the absence of the baseline pathogen after therapy. The adverse event data recorded were the risk of discontinuing due to adverse events, the incidence of serious adverse events, and some common events, such as diarrhea, nausea, headache, constipation, and seizure.

### 2.5. Quality Assessment

Two investigators assessed the risk of bias in each study using the Cochrane Risk-of-Bias Tool 2.0 for RCTs. The Risk of Bias in Non-randomized Studies of Interventions (ROBINS-I) tool was used to evaluate observational studies [12]. The quality of the evidence was ranked based on the risk of bias according to the Grading of Recommendations Assessment, Development and Evaluation (GRADE) approach at the outcome level [13,14].

### 2.6. Statistical Analysis

We used Cochrane Review Manager software RevMan 5 to perform statistical analysis. The degree of heterogeneity was evaluated with the Q statistic test and the I^2^ measure was used to assess the degree of statistical heterogeneity. Heterogeneity was defined as significant when the *p*-value was less than 0.10 or I^2^ more than 50%. The results that are documented include the between-study pooled risk ratio (RR) and 95% confidence intervals (CIs) that were calculated for dichotomous outcomes. The significance of the pooled ratios was determined by the Z test, and results with a *p* value of less than 0.05 were considered statistically significant. Funnel and network plots were also generated by Cochrane Review Manager software RevMan 5. A funnel plot associated with therapeutic regimens was used to examine potential publication bias.

## 3. Results

### 3.1. Characteristics of the Included Trials

The flow diagram in Figure 1 shows the details of the study selection process. There were 156 duplicate articles. After excluding duplicates and irrelevant studies, 44 potentially relevant articles remained. After full-text article review, 35 articles were excluded because they lacked results comparing doripenem to other antimicrobial agents in adult pneumonia patients. Finally, nine studies were included in the meta-analysis [6,15,16,17,18,19,20,21,22]. The main characteristics of the nine included studies are shown in Table 1. There were 952 patients in the doripenem group and 1183 patients in the comparator group. Six were RCTs, one was prospective observational study, and two were retrospective observational studies. Nine studies compared doripenem with other antimicrobial agents, including imipenem/cilastatin in five studies, meropenem in four studies, and piperacillin/tazobactam in two studies. Seven studies had high risk of bias (Figure 2 and Table 2).

### 3.2. Efficacy and Safety Outcomes

Four studies involving 882 patients (449 receiving doripenem therapy, 433 receiving other antimicrobial agent therapies) reported microbiological cure rates. There was no statistically significant difference in the microbiological cure rate between patients treated with doripenem and those treated with other antimicrobial agents (OR = 1.13, 95% CI = 0.83–1.55, *p* = 0.44, I^2^ = 13%) (Figure 3). A total of 7 studies involving 1738 patients (863 receiving doripenem therapy, 875 receiving other antimicrobial agent therapies) reported clinical cure rates. There was no statistically significant difference in the clinical cure rate between patients treated with doripenem and those treated with other antimicrobial agents (OR = 1.01, 95% CI = 0.95–1.07, *p* = 0.74, I^2^ = 0%) (Figure 4). Six studies involving 1952 patients (865 receiving doripenem therapy, 1087 receiving other antimicrobial agent therapies) reported all-cause mortality. There was no statistically significant difference in all-cause mortality between patients treated with doripenem and those treated with other antimicrobial agents (OR = 1.11, 95% CI = 0.85–1.45, *p* = 0.43, I^2^ = 0%) (Figure 5). Three studies involving 1186 patients (596 receiving doripenem therapy, 590 receiving other antimicrobial agent therapies) reported adverse events. There was no statistically significant difference in adverse events between patients treated with doripenem and those treated with other antimicrobial agents (OR = 0.97, 95% CI = 0.73–1.30, *p* = 0.84, I^2^ = 0%) (Figure 6).

Three studies involving 77 patients (42 receiving doripenem therapy, 35 receiving other antimicrobial agent therapies) reported PA pneumonia microbiological cure rates. There was no statistically significant difference in the PA pneumonia microbiological cure rate between patients treated with doripenem and those treated with other antimicrobial agents (OR = 1.98, 95% CI = 0.77–5.10, *p* = 0.16, I^2^ = 41%) (Figure 7). A total of 3 studies involving 149 patients (69 receiving doripenem therapy, 80 receiving other antimicrobial agent therapies) reported PA pneumonia clinical cure rates. There was no statistically significant difference in the PA pneumonia clinical cure rate between patients treated with doripenem and those treated with other antimicrobial agents (OR = 1.38, 95% CI = 0.72–2.67, *p* = 0.34, I^2^ = 59%) (Figure 8). A total of 2 studies involving 115 patients (49 receiving doripenem therapy, 66 receiving other antimicrobial agent therapies) reported PA pneumonia all-cause mortality. There was no statistically significant difference in PA pneumonia all-cause mortality between patients treated with doripenem and those treated with other antimicrobial agents (OR = 0.83, 95% CI = 0.36–1.91, *p* = 0.67, I^2^ = 77%) (Figure 9).

## 4. Discussion

The current meta-analysis of nine studies provides evidence that doripenem has a microbiological cure rate, a clinical cure rate, an all-cause mortality, and adverse events similar to those of comparator antimicrobial agents for the treatment of VAP and HAP. In terms of PA pneumonia, doripenem has a microbiological cure rate, a clinical cure rate, and an all-cause mortality similar to those of comparator antimicrobial agents for the treatment of PA pneumonia. There were only two meta-analyses reported in the literature that explored doripenem therapy for bacterial infections [23,24]. The two meta-analyses also reported that doripenem had a microbiological cure rate, a clinical cure rate, and an all-cause mortality similar to those of comparator antimicrobial agent therapies for bacterial infections. Only three references assessing nosocomial pneumonia were cited in the two meta-analyses [6,17,18]. Our current meta-analysis focused on nosocomial pneumonia and cited nine references. Our study provides more reliable evidence of doripenem efficacy in patients with nosocomial pneumonia.

Regarding the microbiological cure rate, four of the nine studies reported no statistically significant difference in the microbiological cure rate of nosocomial pneumonia between the doripenem and the comparator therapy groups. The five other studies did not compare microbiological cure rates between the two groups. Regarding the clinical cure rate, six of the nine studies reported no statistically significant difference in the nosocomial pneumonia clinical cure rate between the doripenem and the comparator therapy groups. One study did not compare the clinical cure rate between the two groups. One study compared the PA pneumonia clinical cure rate between the two groups. One study reported that the clinical cure rate was lower in patients in the doripenem arm than in those in the imipenem arm (36/79 = 45.6% versus 50/88 = 56.8%; 95% CI, −26.3% to 3.8%). There was no statistically significant difference between the two groups (*p* = 0.165 by Fisher’s exact test). Regarding all-cause mortality, five of the nine studies reported no statistically significant difference in the all-cause mortality of nosocomial pneumonia between the doripenem and the comparator therapy groups. Two studies did not compare all-cause mortality between the two groups. One study compared PA pneumonia all-cause mortality between the two groups. One study reported that all-cause mortality was higher in patients in the doripenem arm than in those in the imipenem arm (17/79 = 21.5% versus 13/88 = 14.8%; 95% CI, −5.0 to 18.5). There was no statistically significant difference between the two groups (*p* = 0.314 by Fisher’s exact test). Regarding adverse events, three of the nine studies reported no statistically significant difference in adverse events between the doripenem and the comparator therapy groups. Six other studies did not compare adverse events between the two groups.

Regarding the PA pneumonia microbiological cure rate, three of the nine studies reported no statistically significant difference in the microbiological cure rate of nosocomial pneumonia between the doripenem and the comparator therapy groups. Six other studies did not compare the microbiological cure rate between the two groups. Regarding the PA pneumonia clinical cure rate, two of the nine studies reported no statistically significant difference in the clinical cure rate of nosocomial pneumonia between the doripenem and the comparator therapy groups. Six studies did not compare the clinical cure rate between the two groups. One study reported that the PA pneumonia clinical cure rate was lower in patients with *Pseudomonas aeruginosa* VAP in the doripenem arm than in the imipenem arm (7/17 = 41.2% versus 6/10 = 60.0%; 95% CI, −57.2 to 19.5). There was no statistically significant difference between the two groups (*p* = 0.440 by Fisher’s exact test). Regarding PA pneumonia all-cause mortality, one of the nine studies reported no statistically significant difference in all-cause mortality in nosocomial pneumonia patients between the doripenem and the comparator therapy groups. Seven studies did not compare all-cause mortality between the two groups. One study reported that PA pneumonia all-cause mortality was higher in patients in the doripenem arm than in those in the imipenem arm (6/17 = 35.3% versus 0/10 = 0.0%; 95% CI, 12.6 to 58.0). There was no statistically significant difference between the two groups (*p* = 0.057 by Fisher’s exact test). The analysis of PA pneumonia was limited by a small sample size and heterogeneity among the studies. Therefore, the evidence related to PA pneumonia was considered low quality in the current meta-analysis.

Kollef et al.’s study showed that there was no statistically significant difference in the clinical cure rate and all-cause mortality between the doripenem and the imipenem subgroups in the VAP patient group. There was also no statistically significant difference in the clinical cure rate and all-cause mortality between the doripenem therapy and imipenem therapy subgroups in the *Pseudomonas aeruginosa* VAP group [6]. The Infectious Diseases Society of America (IDSA) and the American Thoracic Society recommend the following: VAP patients should receive a seven-day course of antimicrobial therapy rather than a longer duration. There may be situations in which a shorter or a longer duration of antibiotics is indicated, depending upon the rates of improvements in clinical, radiologic, and laboratory parameters [25]. A fixed seven-day course of doripenem therapy for VAP does not depend upon the rates of improvements in clinical, radiologic, and laboratory parameters; thus, the antibiotic course does not adhere to clinical treatment norms. Kollef noted one important limitation of their study. There were larger numbers of cases of VAP attributed to *Pseudomonas aeruginosa*, *Acinetobacter baumannii*, and Methicillin-resistant *Staphylococcus aureus* (MRSA) infection in the doripenem arm than in the imipenem/cilastatin arm. The clinical outcome of VAP treated with a seven-day course of doripenem should be worse than that of VAP treated with a ten-day course of imipenem/cilastatin. Medical experts should re-evaluate the clinical validity of Kollef et al.’s study. However, the outcomes of the current meta-analysis of nosocomial pneumonia treatments showed that doripenem was as effective as other antimicrobial agents.

## 5. Limitations

Few prospective RCTs have explored this issue. We included the findings of observational studies in the current meta-analysis. Selection bias and confounding were impossible to eliminate. In addition, the numbers of included studies with certain comparisons were small. There is insufficient data for analysis on the treatment of Pseudomonas aeruginosa pneumonia, which was a limitation of this meta-analysis. A pharmacokinetic/pharmacodynamic evaluation would help to verify the effectiveness of doripenem therapy for nosocomial pneumonia. We did not explore this issue in the current meta-analysis, which was another limitation. There was low quality evidence in the current meta-analysis. However, the conclusions of this meta-analysis are similar to the conclusions of most studies in the literature.

## 6. Conclusions

The current meta-analysis displayed that doripenem treatment for nosocomial pneumonia was associated with a similar microbiological cure rate, clinical cure rate, all-cause mortality, and similar adverse events to comparator antimicrobial agents. However, the evidence in the current meta-analysis was of low quality, and RCTs are urgently needed to confirm the role of doripenem therapy for nosocomial pneumonia.

## Figures and Tables

**Figure 1 jcm-11-04014-f001:**
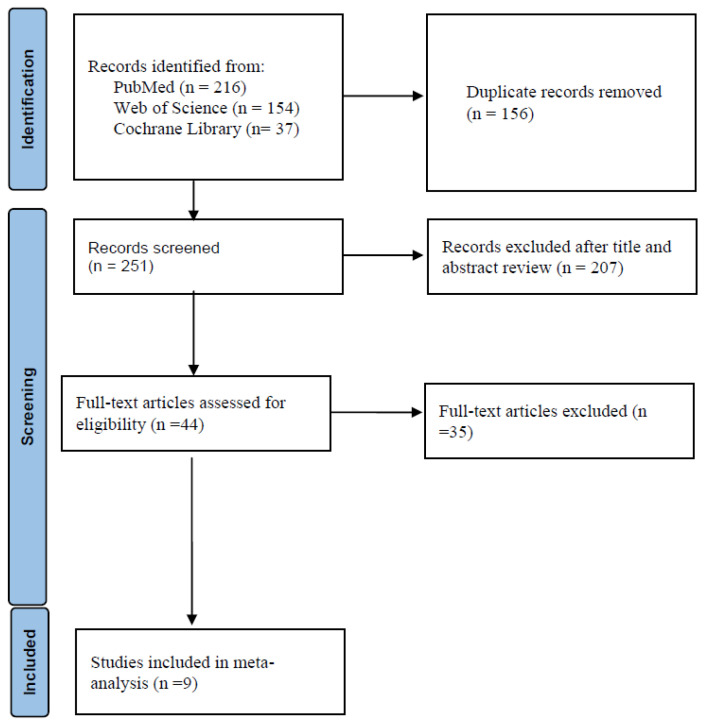
Flow diagram of the study selection process.

**Figure 2 jcm-11-04014-f002:**
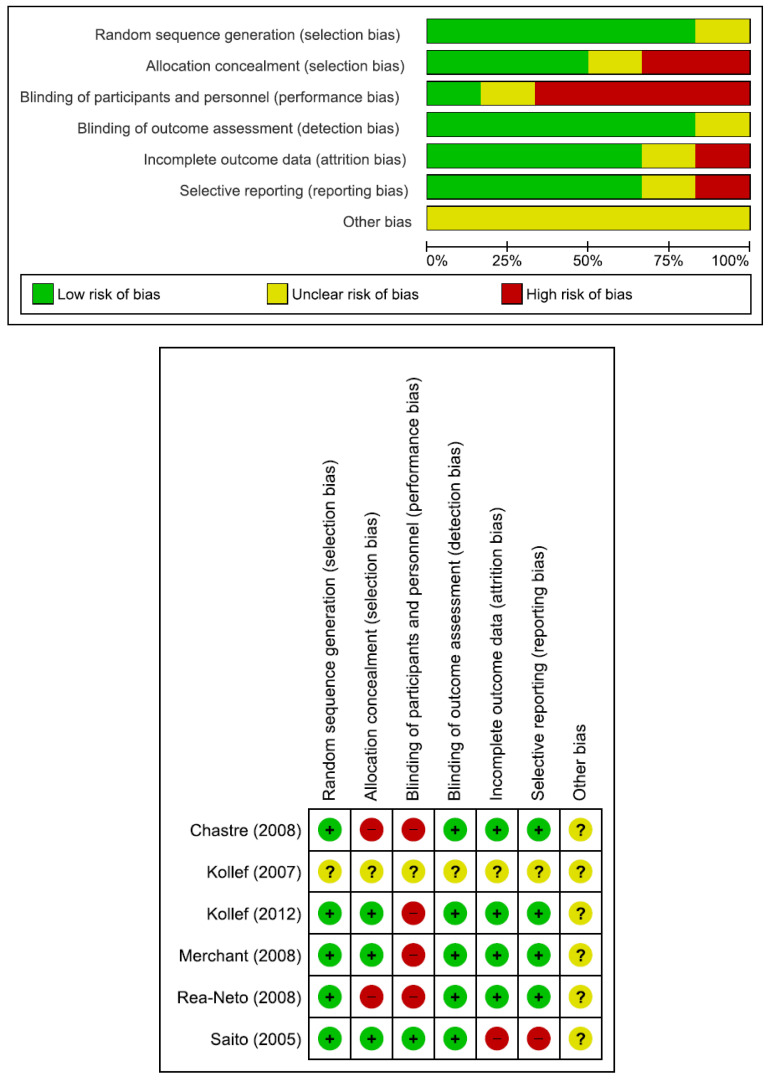
Risk of bias of six randomized controlled trials included in the meta-analysis [6,15,16,17,18,19].

**Figure 3 jcm-11-04014-f003:**
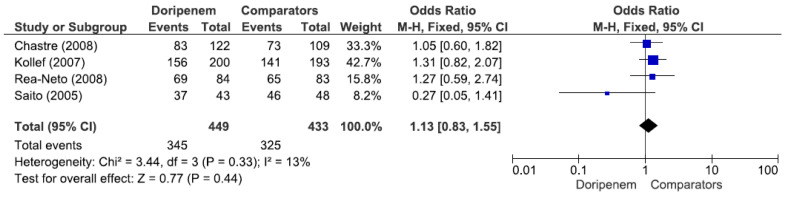
Microbiological cure rate between doripenem and comparator antimicrobial agents in nosocomial pneumonia patients. Four studies involving 882 patients (449 receiving doripenem therapy, 433 receiving other antimicrobial agent therapy) reported microbiological cure rates [15,16,17,18].

**Figure 4 jcm-11-04014-f004:**
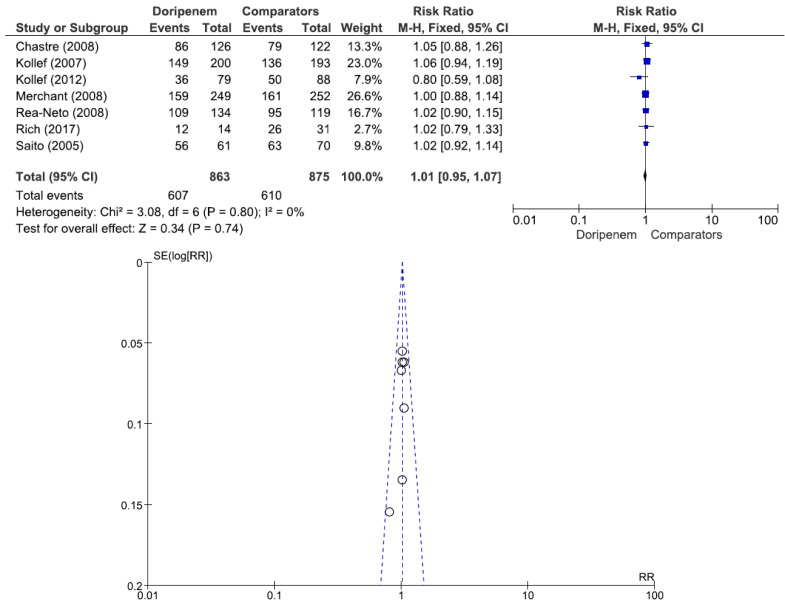
Clinical cure rate between doripenem and comparator antimicrobial agents in nosocomial pneumonia patients. In total, 7 studies involving 1738 patients (863 receiving doripenem therapy, 875 receiving other antimicrobial agent therapy) reported clinical cure rates [6,15,16,17,18,19,21].

**Figure 5 jcm-11-04014-f005:**
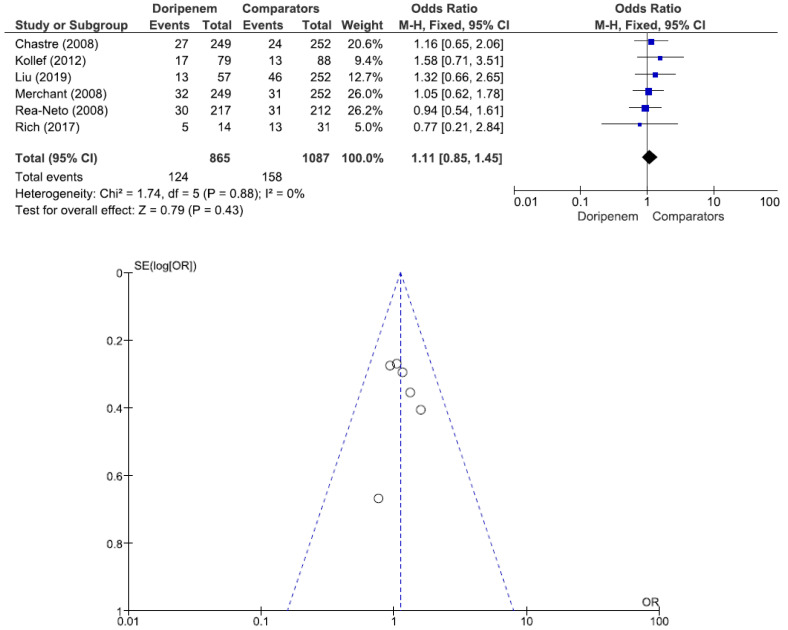
All-cause mortality between doripenem and comparator antimicrobial agents in nosocomial pneumonia patients. Six studies involving 1952 patients (865 receiving doripenem therapy, 1087 receiving other antimicrobial agent therapies) reported all-cause mortality [6,17,18,19,21,22].

**Figure 6 jcm-11-04014-f006:**
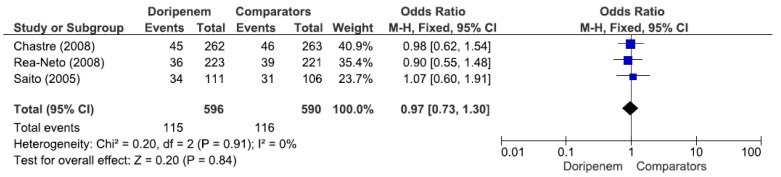
Adverse events between doripenem and comparator antimicrobial agents in nosocomial pneumonia patients. In total, 3 studies involving 1186 patients (596 receiving doripenem therapy, 590 receiving other antimicrobial agent therapies) reported adverse events [15,17,18].

**Figure 7 jcm-11-04014-f007:**
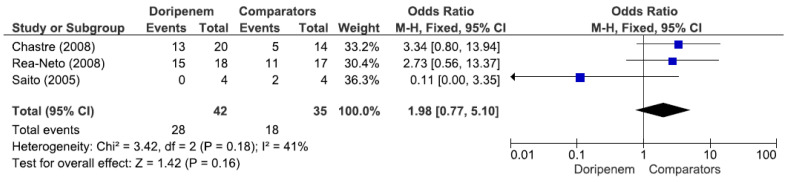
Microbiological cure rate between doripenem and comparator antimicrobial agents in *Pseudomonas aeruginosa* pneumonia patients. A total of 3 studies involving 77 patients (42 receiving doripenem therapy, 35 receiving other antimicrobial agent therapies) reported PA pneumonia microbiological cure rates [15,17,18].

**Figure 8 jcm-11-04014-f008:**
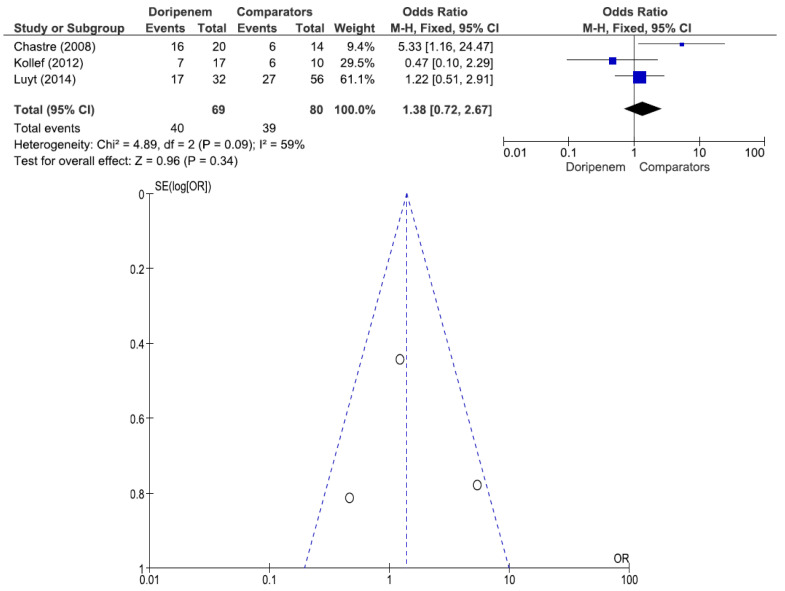
Clinical cure rate between doripenem and comparator antimicrobial agents in *Pseudomonas aeruginosa* pneumonia patients. A total of 3 studies involving 149 patients (69 receiving doripenem therapy, 80 receiving other antimicrobial agent therapies) reported PA pneumonia clinical cure rates [16,18,20].

**Figure 9 jcm-11-04014-f009:**
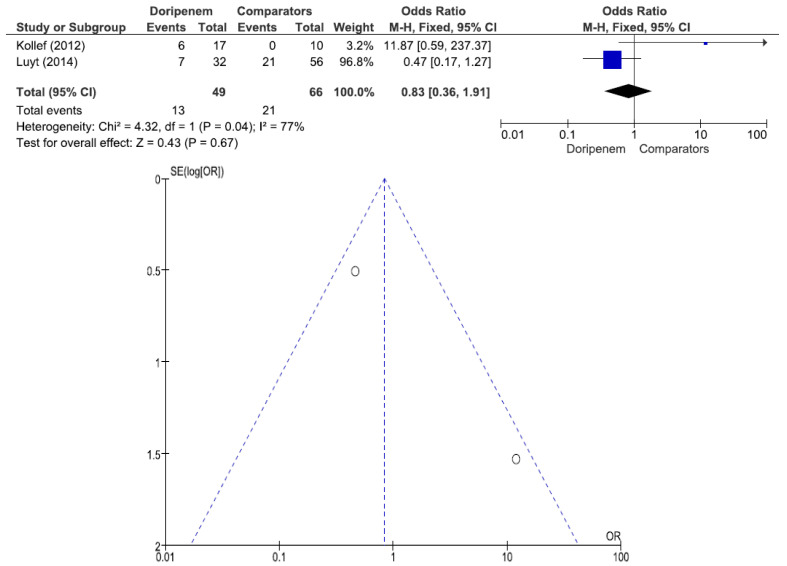
All-cause mortality between doripenem and comparator antimicrobial agents in *Pseudomonas aeruginosa* pneumonia patients. A total of 2 studies involving 115 patients (49 receiving doripenem therapy, 66 receiving other antimicrobial agent therapies) reported PA pneumonia all-cause mortality [16,20].

**Table 1 jcm-11-04014-t001:** Characteristics of nine studies included in the meta-analysis.

Author/Year	Country	Study Design	Pneumonia	Comparators	Drug Dosage	Duration of Therapy
Saito, A., 2005 [15]	Japan	RCTM	pneumonia	MPM	DOR: 250 mg q12h MPM:500 mg q12h	7 days
Kollef, M.H., 2007 [16]	USA	RCTM	HAP, VAP	IMI, TZA	No data	No data
Rea-Neto, A., 2008 [17]	USA, Brazil	RCTM	HAP, VAP	TZA	DOR: 500 mg q8h TZA: 4.5 gm q6h	7–14 days
Chastre, J., 2008 [18]	USA France	RCTM	VAP	IMI	DOR: 500 mg q8h IMI: 500 mg q6h	7–14 days
Merchant, S., 2008 [19]	USA	RCTM	VAP	IMI	DOR: 500 mg q8h IMI: 500 mh q6h or 1.0 gm q8h	7–14 days
Kollef, M.H., 2012 [6]	USA France	RCTM	VAP	IMI	DOR: 1.0 gm q8h IMI: 1.0 gm q8h	DOR: 7 days IMI: 10 days
Luyt, C.E., 2014 [20]	France	PRO	PA-VAP	IMI, MPM	DOR: 500 mg q8h IMI: 1.0 gm q8h MPM: 1.0 gm q8h	More than 5 days
Rich, R., 2017 [21]	USA	RET	HAP, VAP	MPM	DOR: 500 mg q8h MEM: 1.0 gm q8h	7 days
Liu, W.D., 2019 [21]	Taiwan	RET	HAP, VAP	MPM	DOR: 500 mg q8h MEM: 1.0 gm q8h	No data

Foot notes: RCTM: multicenter randomized controlled trial; PRO: prospective study; RET: retrospective study; HAP: hospital-acquired pneumonia; VAP: ventilator-associated pneumonia; PA: *Pseudomonas aeruginosa*; DOR: doripenem; TZA: piperacillin-tazobactam; IMI: imipenem; MPM: meropenem. USA: United states of America.

**Table 2 jcm-11-04014-t002:** Risk bias of three observational studies included in the meta-analysis.

Author	Confounding	Selection	Interventions Classification	Interventions Deviations	Missing Data	Measurement of Outcomes	Selective Results
Luyt, C.E., 2014 [20]	Low risk	Low risk	Low risk	Moderate risk	Low risk	Low risk	Low risk
Rich, R., 2017 [21]	Unclear	Unclear	Unclear	Unclear	Unclear	Unclear	Unclear
Liu, W.D., 2019 [22]	High risk	Serious risk	Serious risk	Serious risk	High risk	Moderate risk	Moderate risk

## Data Availability

Not applicable.

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
