# Peer review of "Doripenem in the Treatment of Patients with Nosocomial Pneumonia: A Meta-Analysis"

_jcm, 2022, doi:10.3390/jcm11144014_

Round 1
Reviewer 1 Report
Dear author, The study on the efficacy and safety of the use of doripenem in patients with nasocomial pneumonia is interesting, as it searched the literature for the best evidence on the subject and tried to clarify the associated results. However, it misses some important details that deserve further clarification.
For example, regarding outcomes, nothing is explained about definitions of these outcomes. How were adverse events defined? What was considered clinical cure? Is the mortality in hospital?
Below are the considerations in detail:
Majors comments
Abstract- The inclusion of the discussion in this item is a little unusual.
Still in this item, the methods could include inclusion and exclusion criteria briefly as well as the period.
Could you also describe quantitative data in the results?
Introduction- Could you at the end of this item include a hypothesis? Because this information clarifies which better outcomes would be needed for this study.
Methods- The searched terms were poorly described, could it be improved?
Did the present meta-analysis include patients with renal failure of any severity? were patients in circulatory shock also involved? Who were patients include?
Was the selection of articles carried out by only two researchers? How was the impartiality of the selection of articles guaranteed?
The tool used for the risk of bias seems to me not suitable for evaluating observational studies, so how was this addressed in these studies?
The biggest weakness of this study was the lack of clear definition of primary and secondary outcomes. Could you clarify?
PK/PD evaluation would be interesting to verify the effectiveness of the medication. I think this issue is a limitation in this study.
Results- Meta-analyses of RCTs with observational studies is unusual and may decrease the quality of results. Why do it then?
Could you describe how many patients remained in the nine studies? And how many in the intervention group and how many in the control group in general?
The related studies to evaluate the treatment of pseudomonas seemed to me to be quite heterogeneous, the I2 value is high. For this reason, that verification is weakened to reach an adequate conclusion. So, it should be further discussed.
Discussion- Some limitations should be discussed with more emphasis in this manuscript, such as the selection of articles, clinical studies analyzed with prospective and retrospective observational studies that may impact on quality and insufficient data for a more conclusive analysis on the treatment of pseudomonas. In addition, it does not present the definitions of outcomes more clearly.
Conclusions- Could you please remove the sentença " Doripenem was not inferior to other antibacterial agents in terms of efficacy and safety"?
Minor comments
References do not seem to have the same pattern, for example some have DOI and others do not.
Reviewer 2 Report
This is a reasonably structured manuscript without a major flaw. The number of studies included is low, especially in Figure6-9. Authors report no significance in any of the comparisons but it would be interesting to re-exam statistical significance when more studies are included in the future.
Author Response
1.Thank you for your kind comment on my article and it helps me a lot.
2.Few prospective RCTs have explored this issue. We included the findings of observational studies in the current meta-analysis. Selection bias and confounding were impossible to eliminate. In addition, the numbers of included studies with certain comparisons were small. There is no sufficient data for analysis on the treatment of Pseudomonas aeruginosa pneumonia, which was a limitation of this meta-analysis. However, the evidence in the current meta-analysis was of low quality, and RCTs are urgently needed to confirm the role of doripenem therapy for nosocomial pneumonia.
Round 2
Reviewer 1 Report
After a detailed analysis of the study, I consider the answers sufficient.
I have only one minor comment: It is necessary to check that the references are in accordance with the journal's standard.